# Personalized Federated Hypernetworks for Privacy Preservation in Multi-Task Reinforcement Learning

## Abstract

Multi-Agent Reinforcement Learning currently focuses on implementations where all data and training can be centralized to one machine. But what if local agents are split across multiple tasks, and need to keep data private between each? We develop the first application of Personalized Federated Hypernetworks (PFH) to Reinforcement Learning (RL). We then present a novel application of PFH to few-shot transfer, and demonstrate significant initial increases in learning. PFH has never been demonstrated beyond supervised learning benchmarks, so we apply PFH to an important domain: RL price-setting for energy demand response. We consider a general case across where agents are split across multiple microgrids, wherein energy consumption data must be kept private within each microgrid. Together, our work explores how the fields of personalized federated learning and RL can come together to make learning efficient across multiple tasks while keeping data secure.

## 1 Introduction

As Reinforcement Learning (RL) is brought to bear on pressing societal issues such as the green energy transition, the types of environments that RL must perform well in may display characteristics exotic to classical RL environments. Real applications at scale may require privacy guarantees which are not provided by modern multi-agent RL algorithms as they may train on privileged or corporate data (Lowe et al., 2017; Sunehag et al., 2017; Rashid et al., 2018); any app that personalizes an RL agent to individual users must take care to protect their privacy by not storing all their data in a central server. Real world applications will also likely feature heterogeneous tasks; every user, robot, energy system will have different traits that cannot be accounted for by "one size fits all" algorithms. As previous work in privacy-preserving RL (Qi et al., 2021; Wang et al., 2020c; Ren et al., 2019; Anwar & Raychowdhury, 2021) does not extend to personalized models, the competing goals of privacy and personalization must be accomplished at the other's expense.

One approach toward privacy preservation by decentralizing data servers within *supervised* learning is **federated learning** (Shokri & Shmatikov, 2015). Federated learning algorithms train a global model from gradient updates sent by individual clients training on their own data, which is never sent to the central server. An extension of federated learning technique is personalized federated learning using hypernetworks (PFH, Shamsian et al. (2021)), which allows for behavior tailored to individual heterogeneous tasks by splitting the model into a global common component (i.e. the **hypernetwork**), and a local individual component (a **local network** generated by the hypernetwork), which is tailored to each client. This task specialization allows for learning common features together in the global component while allowing for learning client-specific knowledge in the local component.

We present a novel application of PFH to RL in a realistic power systems setting that requires both privacy and heterogeneity in agents to accommodate diverse, sensitive environments. An RL controller optimizing hourly transactive energy pricing has been shown to optimize energy usage (Li & Hong, 2014; Spangher, 2021; Vázquez-Canteli et al., 2019; Agwan et al., 2021) by incentivizing consumers, at the scale of groups of buildings (microgrids) or office workers within buildings, to shift demand to different times of day. By guiding consumers to defer energy demands to hours

when solar is especially active, it is possible to drop a building's carbon impact to 48% of normal operation through RL price-setting (Jang et al., 2022), which could have massive implications for grid sustainability. However, RL can be extremely data hungry; prior transactive control attempts required about 80 years of training data (Agwan et al., 2021).

To increase the amount of data available, we consider multiple RL agents, each managing their own (slightly different) microgrid through energy prices and collecting data in parallel. This microgrid environment is a multi-task, multi-agent setup in which the management of each microgrid, through prices, constitutes a task. We characterize our problem as multi-agent because we have multiple RL agents optimizing a shared reward (total profit), and multi-task because optimization of profit in each of the different microgrids presents tasks that are related but also independent due to differences in size, number of batteries in each building, etc. We hypothesize we can accelerate training by incorporating data from multiple microgrids with different characteristics. Learning to set prices using data from multiple microgrids (source tasks) also opens the door to few-shot learning in new microgrids (target tasks), wherein we learn to generate near-optimal prices for a microgrid very quickly.

However, energy data is data in which privacy concerns are paramount. It is our hope to contribute to privacy protection by aggregating learning, *not* data, to one central source. Not only would keeping data of buildings' energy consumption at one central location present a major privacy concern if this central machine is compromised, but message passing of the raw data could present an additional source of vulnerability. Although each microgrid might have access to the data of a few buildings at a time, the scale of damage would be much larger if data was stored in a central server across multiple microgrids.

We now present a hypothetical setting in which our architecture would be useful. One could imagine a hacker being able to learn when the hypothetical company CovertAI trains their new 80 quintillion parameter language model CPT-4 from the energy consumption of CovertAI's compute warehouses. The hacker could sabotage power lines at the right moment to erase learning gains. They may then turn their attention to residential neighborhoods. Here, they could figure out when people are not home from the energy consumption of domestic buildings, timing a theft; they could also disaggregate energy signals to learn the appliances the homeowner has or glean sensitive health information if medical devices produce noticeable patterns in energy consumption.

Applying PFH to the energy application remedies both of these competing issues. PFH takes privacy-preservation into account by design, and accounts for heterogeneous tasks by generating RL agents individualized to each microgrid's size, number of solar panels, batteries, etc. We demonstrate that PFH learns the underlying factors that define an environment by applying PFH to the microgrid price-setting problem, where we observe increases on the scale of millions of dollars in total microgrid profit (reward) over federated and local learning. We also demonstrate how PFH can be used for few-shot transfer learning for new local agents entering the system by reporting drastic training speed-ups ($>$100x) when transferring from source tasks to target tasks. Thus PFH drastically increases the feasibility of RL for energy price-setting.

Methodologically, our paper is novel in its presentation of an adaptation of a state of the art privacy-preserving algorithm to RL. To our knowledge[1], we are the first to explicitly apply personalized federated learning to multi-task, multi-agent RL when centralized learning and joint action-values are unavailable. Application-wise, our paper is also novel in its improvement in energy demand response across heterogeneous microgrids. We hope our work highlights an important microgrid environment to the RL community, helps establish the use of PFH within RL, and allows for RL to address problems where learning speed and privacy are fundamental.

## 2 DEFINITIONS

**Energy Demand Response** is a technique used by grid operators to incentivize consumers to shift demand to times when it is better for grid stability/climate emissions, such as when solar energy peaks. Demand response has the same function as grid-level batteries would in easing the volatility of wind and solar energy and is seen as an important tool in the energy transition (Albadi & El-Saadany, 2007).

---

[1]See Appendix A for a discussion of related work.

A **microgrid** is defined as a small group of buildings that transacts energy with each other through some energy aggregator, governed by an hourly energy pricing scheme. One may imagine they are situated close together with respect to not only geography but also the wiring topology of the grid, making trading within the microgrid preferable to trading with the grid. We will refer to groups of microgrids as "**microgrid clusters**". [2]

A **prosumer** is an entity that both consumes and produces energy, like a building with rooftop solar.

We wish to disambiguate between **multi-task** and **multi-agent** for the reader's convenience. We use them in the conventional sense: multi-task relates to multiple, related settings (in our case slightly different MDP's in each different microgrid) whereas multi-agent refers to multiple different policies.

A **hypernetwork** is a neural network that outputs the weights of another neural network.

A **privacy-preserving algorithm** is one that does not require communication or storage of raw data samples to a central server.

## 3 METHODS

### 3.1 LEARNING ENVIRONMENT

The MicrogridLearn (Agwan et al., 2021) environment is an OpenAI Gym (Brockman et al., 2016) environment used to study RL-set pricing in prosumer aggregations. Specifically, the environment is structured such that an RL agent and an energy utility both broadcast a day's worth of hourly buy and sell prices; $\vec{A}_b, \vec{A}_s$, and $\vec{U}_b, \vec{U}_s$, respectively, to a microgrid's simulated prosumers, who choose at the beginning of the day which hours they will transact with the RL agent and which hours they will transact with the energy utility. Each prosumer is an office building composed of a year's worth of historical data and user-defined, non-negative battery and photovoltaic capacities[3] At every step, i.e. one day where all 24 hours are considered, every prosumer solves a convex optimization optimizing their battery charging/discharging, $\vec{u}_+, \vec{u}_-$, to maximize their individual profit; i.e.:

$$\arg\max_{\vec{u}_+, \vec{u}_-} \left[ \langle \max(\vec{A}_s, \vec{U}_s), (\vec{e}_s + \vec{u}_+) \rangle - \langle \min(\vec{A}_b, \vec{U}_s), (\vec{e}_b + \vec{u}_-) \rangle \right] \tag{1}$$

Where $\vec{e}_s, \vec{e}_b$ are inflexible energy generation and consumption, respectively, of the prosumers, $\langle a, b \rangle$ is a dot product, and *the min and max are taken elementwise*. The first term, i.e. the element-wise maximum, is thus the *gross profit* from energy each prosumer sells, and the second term, i.e. the element-wise minimum, is the *gross expenditures* from energy each prosumer buys. Please note that every vector here may be considered a 24 hour vector, and that opposing actions are exclusive (i.e. sell and buy, or charge and discharge.) Thus, entries in the sell vector in which the prosumer is buying are represented by 0's. By ensuring that each prosumer has the ability to transact with either the utility or the microgrid, we incentivize the microgrid to output prices that are better than the utility, guaranteeing a better experience for prosumers under this microgrid structure. One important simplification we have made is that we model human behavior as fixed in $\vec{e}_b$ relative to the price signal; we do not expect humans to change behaviors (e.g., eating lunch at a different time to take advantage of cheaper energy prices). We only model how distributed batteries could be automatically controlled to maximize the prosumer's profit.

In an environment with this collection of prosumers, the RL agent solves an MDP defined by state space $S := (\vec{U}_s, \vec{U}_b, \vec{g}, \vec{e}_{b,t-1}, \vec{e}_{s,t-1}) \in \mathbb{R}^{24+24+24+24+24}$, where $\vec{g}$ is the day's solar prediction[4] and the $\vec{e}_{b/s,t-1}$ are prosumer buying and selling energy from the previous day. The RL controller emits actions $A := (\vec{A}_b, \vec{A}_s) \in \mathbb{R}^{24+24}$. The agent seeks to maximize a long term discounted reward defined by its individual profit, i.e.:

---

[2]Please note that while "microgrid clusters" *does* appear in the literature, there is no clear consensus around the term being the only appropriate term for multiple connected microgrids.

[3]The photovoltaic output is defined by a fixed year's worth of solar generation for one unit of panels, which is then scaled by the number of photovoltaic panels assigned at initialization.

[4]The day's solar predictions are simply read from a csv.

$$\underset{\vec{A}_b, \vec{A}_s}{\arg\max} \left[ \langle \vec{A}_b, \vec{E}_b \rangle - \langle \vec{A}_s, \vec{E}_s \rangle \right] \tag{2}$$

We somewhat abuse the $\vec{E}$ notation to conveniently define $\vec{E}_{b/s}$ as the total amount of energy bought from or sold to the RL agent hour to hour.

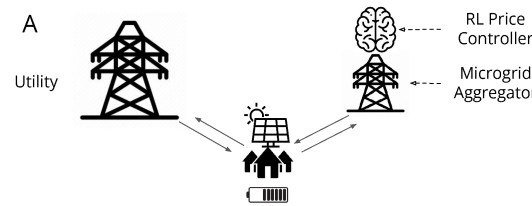

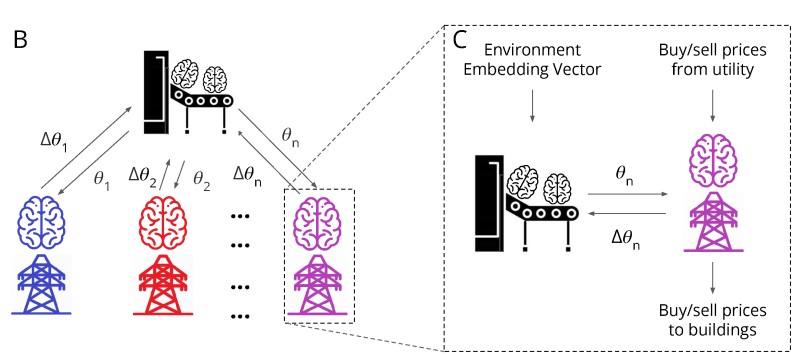

Figure 1: **Microgrids and PFH: A.** We imagine a prosumer that can, at each hour of the day, choose to sell energy surplus or purchase unmet energy demand from the larger utility or to the microgrid aggregator. The microgrid aggregator's energy buy/sell prices are determined by an RL controller. **B.** A Hypernetwork for Personalized Federated Learning (PFH) receives gradient updates from RL controllers and sends back weights. **C.** The hypernetwork takes as input an environment embedding vector and outputs weights for an RL controller. The RL agent takes as input buy/sell prices from the utility and outputs buy/sell prices to the buildings in the microgrid the agent manages. The RL agent sends back a gradient update to the hypernetwork, which uses the update to compute the gradient update for the hypernetwork's own weights.

Because the agent is an aggregator that does not generate its own energy, the profit the agent generates comes from the difference in price between the energy it buys from prosumers at that timestep and the energy it sells to prosumers at that timestep. Any excess supply or demand is transacted with the energy utility. In this way, the environment neatly models a realistic transactive system.

## 3.2 REINFORCEMENT LEARNING

We use Proximal Policy Optimization (Schulman et al., 2017) (PPO), a popular actor-critic based algorithm, to train all of our RL agents to solve the MDP introduced in 3.1 because PPO is reliable and highly performant. Note that both algorithms introduced in 3.3 and 3.4 are agnostic to the architecture of the local policies, so one could use any gradient-based model.

## 3.3 FEDERATED LEARNING

In order to learn a shared model between multiple microgrids in a privacy preserving manner, we turn to federated learning. McMahan et al. (2017) presented what is now the most popular federated learning scheme: Federated Averaging (FedAvg). FedAvg is simple to implement. Denote the parameters for the policy for microgrid $i$ at timestep $t$ as $\theta_{it}$. All the $\theta_{i0}$ are initialized with the same weights, so $\theta_{10} = \theta_{20} = ... = \theta_{n0}$, etc. Then each policy trains on its own microgrid for $k$

local steps, producing a new $\theta'_{ik}$ for each microgrid, which has adapted to be better at price-setting in microgrid $i$ than the original $\theta_{i0}$. All the $\theta'_{ik}$ are transmitted back to a central server, where they compute the shared model for the next iteration by averaging all the $\theta'_{ik}$.

$$\theta_{1k} = \theta_{2k} = ...\theta_{nk} = \frac{1}{n}\sum_{i=1}^{n}\theta'_{ik} \tag{3}$$

Then the local models train on their own, send trained models back to a central server, and repeat. Sending model information only preserves privacy because only the parameters $\theta_{it}$' are communicated with the central server, never any data. Note that in our setup, every client participates in the weight exchange process, not just a sampled subset of the clients. While FedAvg is a simple algorithm that performs well in supervised learning, it learns a global policy for all the price-setting agents. In our case, a global model is not ideal as microgrids may have different energy consumption/supply behaviors.

### 3.4 HYPERNETWORKS FOR PERSONALIZED FEDERATED LEARNING (PFH)

To learn a shared model that is still able to personalize to individual microgrids, we turn to hypernetworks for personalized federated learning. (Shamsian et al., 2021). Personalized federated learning algorithm has found great success in supervised learning, beating FedAvg and personalized federated learning approaches based on meta-learning (Fallah et al., 2020), Moreau Envelopes (T Dinh et al., 2020), and Personalization Layers (Arivazhagan et al., 2019). However, personalized federated learning has never been used before for RL.

Now we will describe PFH more formally. Please refer to Fig. 1 for a visual of the algorithm, and Algorithm 1 for pseudocode. Consider again $\theta_{it} \in \mathbb{R}^m$ as an $m$ dimensional vector denoting the parameters of the policy for microgrid $i$ at timestep $t$. A **hypernetwork** is a neural network that outputs the parameters of another neural network. We will have one global $H_{\phi_t} \in \mathbb{R}^l \to \mathbb{R}^m$ parameterized by $\phi_t$. $H_{\phi_t}$ takes as input an environment embedding vector $v_i \in \mathbb{R}^l$, which is learned for each environment along with the hypernetwork. We initialize $\theta_{i0} = H_{\phi_0}(v_i) \ \forall i \in [1, ..., n]$. Then each[5] local agent trains for $k$ steps, producing new parameters $\theta'_{ik}$. Then, $\Delta\theta_{ik} = \theta'_{ik} - \theta_{i0}$, is sent back to the central server, where it is used to update the hypernetwork:

$$\phi_k = \phi_0 - \frac{1}{n}\sum_{i=1}^{n}\alpha\nabla_{\phi_0}\theta_{i0}^T\Delta\theta_{ik} \tag{4}$$

Since the hypernetwork outputs neural networks conditioned on the environment, it is able to create RL agents that are personalized to the needs of each microgrid. We also still preserve privacy by only communicating parameters with the central server instead of data.

### 3.5 DIVERSITY AND OPTIMAL USE OF PFH

One factor that could affect the relative performance of PFH is the heterogeneity of the scenario. A homogeneous scenario (imagine a cookie-cutter residential neighborhood) could be suitable for federated learning methods due to similarity in behavior. In contrast, an extremely heterogeneous scenario (imagine mixed-use city blocks with night-life, shopping, and residential real estate) could have wildly different energy demands, which may be better learned by individual local networks without any mechanism to share learning. We hypothesize PFH will perform competitively in some average of these two extremes. If local environments are diverse yet share similar underlying mechanisms, PFH will be able to fit to local conditions while sharing information on common trends.

---

[5]Note our setup is slightly different from the original PFH (Shamsian et al., 2021); every client participates in each round, not just a sampled subset of clients. We made this small variation to better understand whether scaling the algorithm to larger numbers of microgrids would be useful.

---

**Algorithm 1** Personalized Federated Hypernetworks

---

**Input:** Environments $E$ and hypernetwork $H_\phi$. For each environment $e \in E$, an RL policy $A(e)$ and hypernetwork-specified parameters $H_\phi(e)$.
**Hyperparameters:** Number of rounds $R$, number of local training steps per hypernetwork update $K$, learning rate $\alpha$.
**for** $r = 1, \ldots, R$ **do**
   **for** environment $e \in E$ **do**
      Get parameters $\tilde{\theta}(e) := \theta(e) := H_\phi(e)$.
      **for** $k = 1, \ldots, K$ **do**
         Collect rollouts $R$ from $e$ using policy $A(e)$ with parameters $\tilde{\theta}(e)$.
         Update $\tilde{\theta}(e)$ using PPO with rollouts $R$.
   Initialize $\bar{\phi}_{\text{update}} := 0$.
   **for** environment $e \in E$ **do**
      $\Delta\theta(e) := \tilde{\theta}(e) - \theta(e)$
      $\bar{\phi}_{\text{update}} := \bar{\phi}_{\text{update}} + \frac{\nabla_\phi \theta(e)^T \Delta\theta(e)}{|E|}$
   Update hypernetwork parameters $\phi = \phi - \alpha \cdot \bar{\phi}_{\text{update}}$.

---

## 4 EXPERIMENTAL SETUP

### 4.1 SIMULATING DIVERSE MICROGRIDS

Because each microgrid is defined by a distribution of photovoltaics and battery sizes, we propose a simple way to tweak the amount of diversity in a system. We sample photovoltaic and battery sizes from normal distributions, changing the variance $\sigma^2$ as the diversity parameter, and round outcomes to the nearest integer[6] We sample from $\mathcal{N}(\mu = 100, \sigma = 10)$ for low diversity cases, $\mathcal{N}(100, 30)$ for medium diversity and $\mathcal{N}(100, 50)$ for high. We note here that we have chosen the low, medium, and high cases such that 95% of samples (i.e. 2 standard deviations around the mean) in the high case hit realistic bounds in the environment; i.e., 0 (an obvious lower bound) and 200[7].

### 4.2 BASELINES

We compare PFH against FedAvg and two other baselines. First, we observe what happens with no RL control at all; the microgrid aggregator outputs prices that are exactly the same as the utility's. We assume buildings choose to meet half their energy demand/surplus with the utility and half with the aggregator. Our second baseline is the approach used in Agwan et al. (2021): training all the local RL controllers with only their own data: no central model or inter-microgrid communication. These two baselines, no RL and local control, are designed to highlight the added value of RL[8] to the task of price-setting for energy demand response in microgrids, and the added value of having some central model that aggregates learning across multiple microgrids, respectively.

For specification on how we selected hyperparameters, please see Appendix E.

---

[6] As we are sampling from a "hyper" distribution to instantiate houses, the means of the distribution are not as important as the variances in instantiating diversity.

[7] 200 is a realistic upper bound in both solar panels and batteries: 200 solar panels would require an area of 60 x 70 ft, which bounds the square footage of many commercial roofs, and 200 batteries would be a realistic upper bound of entities not engaging in commercial grid services.

[8] The most common non-RL methods for microgrid price-setting are iterative pricing methods (IP) (Liu et al., 2017; Wang & Huang, 2016) in which buildings "bargain" with microgrids to reach equilibrium prices. We exclude these baselines because they require *each building* to develop their own demand forecasts. This requirement raises the computational barrier for entry by an order of magnitude. For comparison, if we had 10 microgrids with 10 buildings each, local RL requires training 10 models (10 microgrids), PFH and FedAvg requires 11 (10 microgrids + 1 central model), and IP requires 100 (10 buildings x 10 microgrids). Agwan et al. (2021) also showed RL results in less volatile pricing curves and better performance compared to IP.

Table 1: Cumulative profits above base utility pricing after 10,000 days, in hundred thousands.

| Scenario | PFH | FedAvg | Local Baseline |
|---|---|---|---|
| Simple, 5 agents | 39.23 | **45.75** | 43.72 |
| Simple, 10 agents | 42.11 | 41.65 | **43.18** |
| Simple, 20 agents | 40.85 | 34.52 | **42.82** |
| Medium, 5 agents | **47.50** | 40.95 | 40.12 |
| Medium, 10 agents | **46.89** | 43.82 | 41.73 |
| Medium, 20 agents | **48.22** | 39.38 | 39.66 |
| Complex, 5 agents | 34.77 | 32.66 | **35.60** |
| Complex, 10 agents | 43.01 | 42.08 | **45.24** |
| Complex, 20 agents | **44.39** | 38.70 | 40.78 |

## 4.3 MULTI-TASK TRANSFER

An interesting feature of our hypernetwork-based setup is the potential for multi-task learning and few-shot transfer learning. The optimization problem of setting prices for each microgrid can be viewed as an individual task. Since the hypernetwork should learn some common strategies for each task, we tested whether it can generalize to unseen tasks with little training. To test this hypothesis, we simply take a hypernetwork that has trained for to manage a microgrid cluster with 20 microgrids of medium diversity and train the hypernetwork to manage a *new* microgrid cluster of 20 microgrids with the same level of diversity. By pretraining our hypernetwork on 20 varied source tasks, we hope to encode enough knowledge applicable to the new target tasks to make few-shot transfer learning possible. We will refer to such a pretrained hypernetwork as a Few-Shot PFH.

## 5 RESULTS AND DISCUSSION

### 5.1 PFH ACCELERATES LEARNING IN MEDIUM DIVERSITY MICROGRID CLUSTERS

Fig. 2 shows average daily profit gained by each microgrid in a microgrid cluster with 5, 10, and 20 microgrids, with varying amounts of diversity. The middle column of Fig. 2 shows PFH is more efficient and profitable for a microgrid cluster than a microgrid cluster under a FedAvg or local control scheme. As shown in Table 1, PFH results in up to $8,500,000 of additional cumulative profit after 10,000 days over the local control baseline in a microgrid cluster with 20 microgrids. However, this advantage does not carry over to cases of small or large diversity. For less diverse scenarios, PFH was comparable or less profitable than FedAvg or local control. For more diverse scenarios, local control was generally more profitable. The number of microgrids in microgrid clusters also did not seem to have much effect on learning speed here.

### 5.2 FEDAVG RECOVERS LOCAL PERFORMANCE AT BEST

Curiously, our results indicated that FedAvg presented did not improve the management of a microgrid cluster over a collection of local agents. We had expected FedAvg to perform better in the homogeneous case, and to scale with the number of agents, but neither effect appears in our results. Although FedAvg may perform well in supervised learning (McMahan et al., 2017), it may not extend well to RL. We explain FedAvg's poor performance as follows: unlike supervised learning, RL requires exploration and already suffers from non-IID data. When aggregating learning across different heterogeneous environments, this issue of learning from non-IID data may have been exacerbated, slowing down learning. Furthermore, the setting of federated learning may have made the RL algorithm more sensitive to hyperparameters, as the set of hyperparameters that works for all tasks is likely smaller than the set that works for any one task. We conducted an extensive hyperparameter sweep that is documented in Appendix E to account for this issue with combining federated

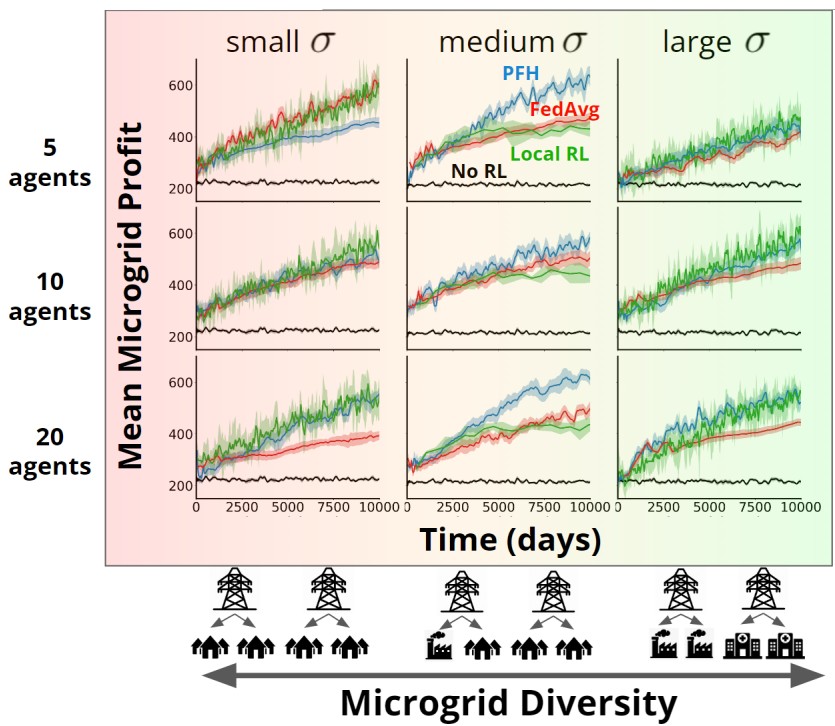

Figure 2: **RL Agent Performance:** The performance of the RL price-setting agent as a function of the number and diversity of the microgrids in the microgrid cluster . Performance is measured by looking at the average daily profit gained by each microgrid.

learning with RL. Meanwhile, the hypernetwork is able to learn how to build RL policies that are less sensitive to these hyperparameters because it outputs agents personalized to each task.

### 5.3 PFH ENABLES FEW-SHOT LEARNING

Fig.3.A shows the hypernetwork adapted to a new set of microgrid management tasks extremely quickly. On average, within $\approx 1.5$ months (42 days), each new microgrid achieved $\approx \$380$ in daily profit, which is about the daily profit of the local agents baseline after 13 years (5000 days) of training. The original, randomly initialized PFH required 3000 days to achieve similar performance. Thus, Few-Shot PFH achieved a 119x speedup over local agents and 71.4x over a randomly initialized PFH over the first 1.5 months. Within 7 months (210 days), Few-Shot PFH achieved a daily profit of $565: 44% higher profit than the local agents ever achieve. A randomly initialized PFH required $\approx 22$ years (8000 days) to achieve similar performance: a 38x speedup in the first 7 months of training. Cumulatively, having a pretrained PFH on 20 microgrids saves $\approx \$1,500,000$ over the course of training on the new microgrid management tasks compared to a randomly initialized PFH.

### 5.4 FEW-SHOT LEARNING CAPABILITY SCALES WITH MICROGRID CLUSTER SIZE

When we tried the same experiment with hypernetworks that were trained for 10,000 days on 5 microgrid management tasks and 10 tasks in Fig. 3.B and tested on 5 and 10, respectively, we saw significantly smaller boosts in the mean reward over groups of new tasks with fewer training tasks. The smaller scale of benefit was expected given a multi-task learning strategy with fewer source tasks and data. Indeed, when trained on 5 tasks, there was hardly any initial training speedup. Starting from 10 tasks, we observed a large initial boost (although not as large as with 20.) Rather strikingly, Few-Shot PFH pretrained on 5 and 10 tasks converged to lower reward curves than even the baseline PFH (i.e., a randomly initialized hypernetwork.) With 20 tasks, we saw both a large initial boost in training speed and no adverse impact on long term training. We hypothesize the fewer microgrid source tasks provided, the more information is stored in the environment embedding, which makes

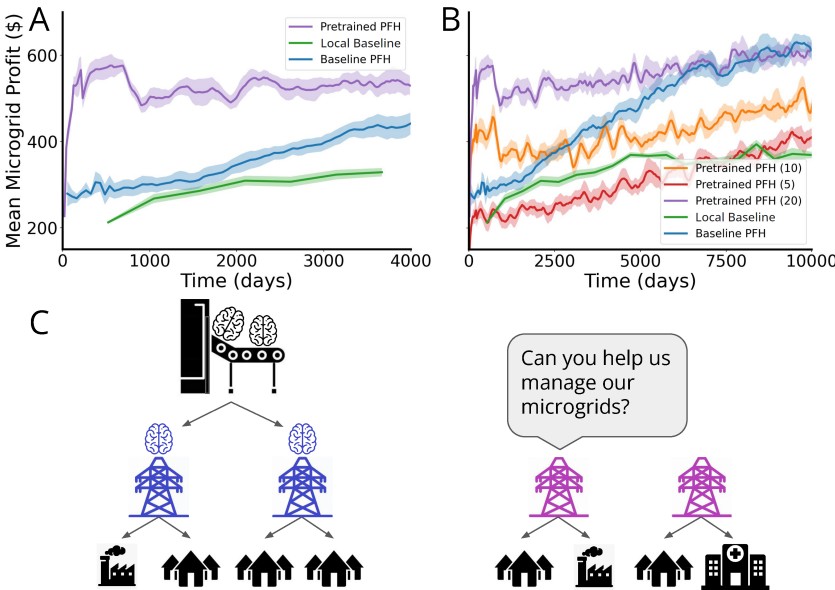

Figure 3: **PFH Enables Few Shot Learning: A.** Mean microgrid profit of PFH pretrained on 20 microgrids learning to manage 20 new microgrids ("Pretrained PFH"), compared to randomly initialized PFH ("Baseline PFH") and the local agents baseline ("Local Baseline"), over training days on the new microgrids. **B.** Mean microgrid profit of PFH pretrained on 5, 10, and 20 microgrids on a new set of microgrids, over a longer time than A. **C.** A plausible scenario in which PFH may need to quickly adapt to new microgrids.

the hypernetwork brittle to new environments. Thus in the 5 and 10 case, the net has not learned enough shared dynamics in the other parameters to generalize to new settings. In the case of 20 and above, we expect that enough shared dynamics are learned that the net can generalize. The range of training speed benefits we observed suggested the potential in some configurations for a Few-Shot PFH to quickly adapt to new tasks depends on how many tasks it was initially trained on.

## 6    LIMITATIONS AND FUTURE WORK

Technically, our work is limited in several ways. We presented a "goldilocks" zone in which PFH outperforms other methods, but as we tested only in simulation, it is unclear where this goldilocks zone would appear in the real world. Second, we protect privacy by only communicating parameters, but it is possible to reconstruct data from parameters for some models (Carlini et al., 2020).

We would like to address these two issues in future work. First, we would like to explore other environments to determine whether the "goldilocks" phenomenon is unique to the MicrogridLearn environment. Second, we would like to combine our PFH training procedure with differential privacy measures like those in Abadi et al. (2016) to further impede reconstruction of training data.

For two more extended examples of future work we are excited to test, as well as further discussion on potential societal impact of the application of our algorithm, please see Appendix C and D.

## 7    CONCLUSION

We seek a privacy preserving mechanism for improved training speeds on profit-driven energy aggregation in a microgrid cluster . To this end, we are the first to demonstrate a PFH for RL to output local model gradient updates and show improved training times. We hypothesize that PFH shines when the setting is diverse enough to differ meaningfully between systems, but not so diverse that system behavior diverges. We prove our hypothesis and demonstrate the efficacy of PFH for few shot learning approaches.

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

## A    RELATED WORK

We position our literature within an ecosystem of work related to transactive pricing in microgrids. A price-setting RL agent was first shown to help an energy aggregator improve demand response and generate a profit (Agwan et al., 2021). Since then a number of works have explored the issue (Shojaeighadikolaei et al., 2021; Wen et al., 2020; Han et al., 2021; Rolnick et al., 2022), with some work exploring different configurations of RL.

We wish to provide an example of federated learning. "Distributed Selective Stochastic Gradient Descent", i.e. DSSGD (Shokri & Shmatikov, 2015), is an interesting example which deserves further exploration from the interested reader. DSSGD has each local model exchange select parameters and

gradient updates with the central server. In contrast, FedAvg (McMahan et al., 2017) just averages all local model gradient updates and syncs all local model parameters. In personalized federated learning, there have been techniques other than PFH to facilitate the federated learning of personalized models such as Moreau envelopes T Dinh et al. (2020), multi-task learning Li et al. (2021), personalization layers Arivazhagan et al. (2019), and local representations Liang et al. (2020). However, Shamsian et al. (2021) compared PFH against several of these algorithms and performed better in supervised learning.

Existing multi-agent environments are often solved through multi-agent RL algorithms like MAD-DPG (Lowe et al., 2017), VDN (Sunehag et al., 2017), and Q-Mix (Rashid et al., 2018), but these aggregate data from all the agents onto one central machine during training, and take advantage of joint action-values from all agents. Other works use federated hypernetworks for multi-task setups, but specifically not those in RL.

The combination of the two fields, federated multi-agent reinforcement learning, has focused mainly on learning global models, not personalized models for heterogeneous tasks Qi et al. (2021); Wang et al. (2020c); Ren et al. (2019); Anwar & Raychowdhury (2021); Kwon et al. (2020); Zhang et al. (2021b); Xu et al. (2021); Wang et al. (2020a); Nadiger et al. (2019). Decentralized multi-agent reinforcement learning does learn personalized models Zhang et al. (2021a; 2018), but it may be difficult to scale up a decentralized system such that each agent can benefit from the experiences of all the others without large communication costs. This is not as much of an issue for federated learning as communication only needs to occur between clients and a server rather than clients and all their peers. Although decentralized systems have their benefits, we focus mainly on federated systems in this work.

We note that "privacy-preservation" might be to an extent an overstatement, as works have shown that the transmission of gradients can allow one to recreate private data (Xie et al., 2019; Hitaj et al., 2017; Melis et al., 2018). Thus, while we note that our work guarantees privacy to the extent of other works within the field of federated learning (Wang et al., 2020b; Li et al., 2019; Acar et al., 2021; Zhang et al., 2020), one should apply the term privacy-preservation with the same caveats to our work as to the rest of the field.

## B    REPRODUCIBILITY STATEMENT

### B.1    OVERVIEW

We wish to provide all details sufficient for the reader to reproduce our experiments. We will overview how to setup the code, which we provide in a zip file for analysis, and discuss hyperparameter selection.

### B.2    CODE SETUP

- First, download and extract the anonymized code provide in the submission.
- Install dvc (with google drive support). On linux this is `pip install 'dvc[gdrive]'`
- Install Docker, if you have not already.
- Run `python3 -m dvc remote add -d gdrive gdrive:// 1qaTn6IYd3cpiyJegDwwEhZ3LwrujK3_x`
- Run `python3 -m dvc pull`
- Run `pip install -r requirements.txt`
- Run `docker build .`

### B.3    HYPERPARAMETER SELECTION

We select hyperparameters for each algorithm by conducting hyperparameter sweeps, with parameters proposed by Bayesian hyperparameter optimization (Seeger, 2004) trying to maximize the mean reward across all agents. For the local agent baseline we run 3 sweeps, one for each level of diversity, as each level has a different distribution of microgrids. For FedAvg and PFH, we conducted 9

sweeps, varying diversity (simple, medium, complex), and the number of agents (5, 10, 20). The number of agents is relevant to FedAvg and PFH because they learn from data across multiple agents. Each local agents and FedAvg sweep had 50 runs, and each PFH sweep had 100 runs, because PFH had ≈ double the number of parameters to optimize.

To minimize the effect of outliers, we use the hyperparameters from the third highest performing run from each sweep. Detailed parameter bounds for hyperparameter sweeps can be found in the appendix in Table 2, and simple analyses of the hyperparameters can be found in Appendix E.2.

## C  MORE FUTURE WORKS

### C.0.1  "COST OF PRIVACY"

We wish to further investigate the "cost" of privacy in terms of the negative impact it may have on training time and thus on cumulative aggregator profit. In order to create a true apples-to-apples comparison, we would need a mechanism that aggregates information across microgrids in a suitable way. Some ideas on this front include multi-agent RL that shares the critic but personalizes policies, and hierarchical RL with a global aggregator.

### C.0.2  VERTICAL INTEGRATION OF THE HIERARCHY

In the future, PFH may enable further exploitation of the hierarchical nature of price setting for energy demand response. The energy grid can be imagined as a hierarchical tree, with buildings responding to energy prices set by microgrids, which respond to energy prices set by city utilities, which respond to energy prices set by state utilities, etc. In the future, we may have IoT devices adjusting demand to energy prices set at the building level. At any level of the energy grid, the task is the same: set prices for agents beneath you to elicit a demand response. In this work we have only looked at one level of this energy hierarchy, but the methods we have used could be applied to other layers of the hierarchy as well, and even multiple levels of the hierarchy. One could imagine a hypernetwork that learns from price-setting agents at every level of the hierarchy, and can be used to rapidly initialize agents to manage any new entrants to the energy grid, all while preserving privacy at different levels of the tree.

## D  DISCUSSION OF SOCIETAL IMPACT

What are potential negative societal effects of our work? Overall, negative effects to prosumers are limited, as the focus of our work is in protecting consumer information. Furthermore, prior work demonstrated that the presence of an aggregator consistently reduced energy costs for consumers.

However, a persistent danger of AI is that it is often deployed through centralized profit-seekers. Our work is no different in this regard. Although our specific innovation protects prosumers, it may improve the economic viability of a profit-seeking entity whose scale may eventually enable it to further its own profit at the expense of prosumers.

Also, the act of setting prices in systems may raise fairness concerns. If initial training microgrids are biased towards wealthier residents, the PFH may initialize new policies with pricing that benefits consumption habits of wealthier clients but not poorer clients. A vivid illustration may be seen in the types of prosumers who are best poised to benefit from economic aggregation: prosumers with large solar panels and batteries are able to shield themselves from or profit off of high prices by consuming their own energy, and may fully charge their batteries when prices are low. Prosumers with smaller or no storage capabilities do not have this luxury, and thus are more vulnerable to the negative effects of price fluctuation.

## E  APPENDIX FOR HYPERPARAMETER EXPLORATION

### E.1  HYPERPARAMETER SWEEP SPECIFICATIONS

See Table 2 for the bounds of the hyperparameter sweeps that we performed.

Table 2: **Hyperparameter Sweep Bounds** All sweeps swept over the first 6 hyperparameters. AFL sweeps additionally swept over the "AFL # of Local Steps" parameter. PFH sweeps additionally swept over the last 7 parameters. Due to the high dimensionality of the sweep, we used Bayesian hyperparameter optimization. We refer to four types of distributions: Uniform, Int Uniform, Log Uniform, and Int Log Uniform. The "Int" type distributions simply quantize the underlying distribution (e.g. if $x$ is sampled from a uniform distribution, $\lfloor x \rfloor$ is returned by an int uniform distribution). The Log Uniform distribution samples uniformly over the log of the value, so the probability of sampling $e^1$ is the same as $e^2$, etc.

| Parameter | Lower Bound | Upper Bound | Distribution |
|---|---|---|---|
| Batch Size | 16 | $\lfloor e^8 \rfloor$ | Int Uniform |
| Learning Rate | $e^{-8}$ | 1 | Log Uniform |
| # of Hidden Layers | 1 | 7 | Int Uniform |
| # of Neurons per Hidden Layer | 1 | $\lfloor e^7 \rfloor$ | Int Log Uniform |
| PPO Clipping Param | 0.01 | 1.0 | Uniform |
| PPO # of SGD Iterations | 1 | 30 | Int Uniform |
| AFL # of Local Steps | 2 | $\lfloor e^6 \rfloor$ | Int Log Uniform |
| PFH Dropout | $e^-10$ | 1 | Log Uniform |
| PFH Embedding Dim | 1 | 512 | Int Uniform |
| PFH L2 Regularization | $e^{-10}$ | 1 | Log Uniform |
| PFH Learning Rate | $e^{-4}$ | 1 | Log Uniform |
| PFH # of Hidden Layers | 1 | 6 | Int Uniform |
| PFH # of Neurons per Hidden Layer | 1 | 1024 | Int Uniform |
| PFH # of Local Steps | 1 | 100 | Int Uniform |

## E.2 REGRESSION EXPLORATIONS OF HYPERPARAMETER SWEEPS

We present, for the reader's interest, a regression fit on the hyperparameters that were swept over. In this regression, each observation is a single run of the sweep, the dependent variable in both is the reward mean of the learning trajectory, and the independent variables are those listed in the rows. We believe that this regression contains some interesting information; specifically on the direction of coefficients (i.e. whether they are negative or positive) and on which parameters were significant in producing a positive reward. We note that the basic assumption of linear regression, that observations are sampled IID from a distribution, is not the case here; observations are loosely dependent on each other as the parameter configurations in each batch are determined by the performance of parameters in the previous batch. Thus, we relegate these results as a curiosity for supplementary material only. We are more confident in the *negative* results of this regression, i.e. which variables are insignificant after controlling for the others, than the positive results, as this indicates parameters that the sweep chose not to focus on. We believe that further work in regressions of hyperparameter values may be an interesting research endeavor for understanding ML models as well as for ML applications like AutoML.

### E.2.1 FULL REGRESSION MODEL (TABLE 3)

Of specific interest in the full regression are which hyperparameters did and did not effect the average reward. Many variables in the hypernetwork itself do not seem to matter: the hypernetwork's learning rate, number of layers, and L2 regularization did not matter. However, whether or not the hypernetwork selected for dropout *did* matter, and it hurt the performance, implying that hypernetwork fitting was more important than robustness. Some parameters of the PPO agents, such as the clip parameter or number of gradient updates, did not matter much either.

### E.2.2 REDUCED REGRESSION MODEL WITH ONLY SIGNIFICANT COEFFICIENTS INCLUDED (TABLE 4)

We believe that this test is more interesting for examining the direction of significant variables after controlling for the other variables. Here, it is interesting to note greater sizes of policy networks, ("sizes"), has a negative effect, while number of layers of policy networks have a positive effect, offering a mixed view on whether policy complexity is important. The learning rate is negatively important, implying that more stability in network is preferred. The number of local model update steps allowed is positively correlated to mean reward, implying that the more the local models are allowed to fit, the better. The combination of a negative effect of batch size and positive effect of learning rate implies that the local RL agents found it easier to take few steps (lower batch sizes) but update policies at a more conservative rate (lower learning rates), which makes sense in the RL context.

Table 3: Full regression model

| Dep. Variable: | Avg Reward | R-squared: | 0.362 |
| Model: | OLS | Adj. R-squared: | 0.348 |
| Method: | Least Squares | F-statistic: | 24.62 |
| Date: | Thu, 19 May 2022 | Prob (F-statistic): | 1.08e-43 |
| Time: | 08:16:34 | Log-Likelihood: | -3188.0 |
| No. Observations: | 533 | AIC: | 6402. |
| Df Residuals: | 520 | BIC: | 6458. |
| Df Model: | 12 | | |

| | coef | std err | t | P> \|t\| | [0.025 | 0.975] |
|---|---|---|---|---|---|---|
| Intercept | 344.6188 | 29.679 | 11.612 | 0.000 | 286.314 | 402.924 |
| sizes | -1.9453 | 0.777 | -2.504 | **0.013** | -3.472 | -0.419 |
| n_layers | 5.2696 | 2.806 | 1.878 | **0.061** | -0.243 | 10.782 |
| learning_rate | -1835.5718 | 128.380 | -14.298 | **0.000** | -2087.779 | -1583.364 |
| ppo_clip_param | 49.4402 | 38.009 | 1.301 | 0.194 | -25.230 | 124.110 |
| hnet_lr | 77.4179 | 63.747 | 1.214 | 0.225 | -47.815 | 202.651 |
| hnet_num_local_steps | 1.0579 | 0.150 | 7.043 | **0.000** | 0.763 | 1.353 |
| ppo_num_sgd_iter | -0.3889 | 0.741 | -0.525 | 0.600 | -1.845 | 1.068 |
| hnet_num_layers | -2.3439 | 2.571 | -0.912 | 0.362 | -7.395 | 2.708 |
| batch_size | -0.6685 | 0.171 | -3.910 | **0.000** | -1.004 | -0.333 |
| hnet_embedding_dim | 0.0077 | 0.028 | 0.271 | 0.786 | -0.048 | 0.063 |
| hnet_l2_reg | -10.4826 | 19.912 | -0.526 | 0.599 | -49.601 | 28.636 |
| hnet_dropout | -68.5849 | 21.070 | -3.255 | **0.001** | -109.978 | -27.192 |

| Omnibus: | 16.401 | Durbin-Watson: | 1.670 |
|---|---|---|---|
| Prob(Omnibus): | 0.000 | Jarque-Bera (JB): | 19.555 |
| Skew: | 0.336 | Prob(JB): | 5.67e-05 |
| Kurtosis: | 3.655 | Cond. No. | 9.08e+03 |

Table 4: Reduced regression model

| Dep. Variable: | Avg Reward | R-squared: | 0.357 |
| Model: | OLS | Adj. R-squared: | 0.349 |
| Method: | Least Squares | F-statistic: | 48.63 |
| Prob (F-statistic): | 1.79e-47 | Log-Likelihood: | -3190.3 |
| No. Observations: | 533 | AIC: | 6395. |
| Df Residuals: | 526 | BIC: | 6425. |
| Df Model: | 6 | | |

| | coef | std err | t | P> \|t\| | [0.025 | 0.975] |
|---|---|---|---|---|---|---|
| Intercept | 364.7212 | 15.515 | 23.507 | 0.000 | 334.242 | 395.201 |
| sizes | -2.1970 | 0.627 | -3.502 | 0.001 | -3.429 | -0.965 |
| n_layers | 5.1216 | 2.759 | 1.856 | 0.064 | -0.299 | 10.542 |
| learning_rate | -1829.2013 | 127.609 | -14.334 | 0.000 | -2079.887 | -1578.516 |
| hnet_num_local_steps | 1.0482 | 0.149 | 7.055 | 0.000 | 0.756 | 1.340 |
| batch_size | -0.6409 | 0.167 | -3.841 | 0.000 | -0.969 | -0.313 |
| hnet_dropout | -65.1701 | 20.923 | -3.115 | 0.002 | -106.273 | -24.067 |

| Omnibus: | 17.230 | Durbin-Watson: | 1.669 |
|---|---|---|---|
| Prob(Omnibus): | 0.000 | Jarque-Bera (JB): | 19.988 |
| Skew: | 0.360 | Prob(JB): | 4.57e-05 |
| Kurtosis: | 3.618 | Cond. No. | 2.57e+03 |

