# OpenReview forum: "Personalized Federated Hypernetworks for Privacy Preservation in Multi-Task Reinforcement Learning"
_ICLR.cc/2023/Conference — Submitted to ICLR 2023_

### Official Review · Reviewer_C5ih · 2022-10-22

**Confidence:** 4
**Correctness:** 4
**Technical Novelty And Significance:** 2
**Empirical Novelty And Significance:** 1
**Recommendation:** 3

**Clarity, Quality, Novelty And Reproducibility:**

Many parts of experiment need additional information to clarity since it is based on a simplified hypothetical simulation. Since it is an application paper, the novelty is more valued on the contribution to solve the application problem. In this paper, the augment of advantage of using this algorithm in this application is not strong. Many critical paper settings were not explained here, which made to reproduce the same experiment challenging.

**Strength And Weaknesses:**

Strength:
The problem formation is clear. The real problem of energy demand between different utility node and consumer behavior is complex. Author made some reasonable simplification to scale down the problem into a manageable format that could be analyzed by simulation.

Weaknesses:
The weakness of the paper is experiment part is not strong enough to show advantage of their algorithm. The experiments show your algorithm is not always work, where only a selected scenario it outperformed baseline.
The formation for total profit should be explained more since it is based on the process of utility market and consumer behavior. The constrain of consumer behavior would be stated since it is unrealistically to have all the consumer to cook at non-dinner period or not use AC during the middle of day.


**Summary Of The Paper:**

This work is an application paper that deployed the existing federated learning technologies namely hypernetwork to achieve personalized learning in reinforcement learning across multiple agents. The problem statement is based on a simplified hypothetical situation where multiple communities are powered by renewable energy with battery to assist the grid. With assumption of consumer will adjust their behavior based on the price changes. The training process of system contains two parts, local training, and global aggregation. The global aggregation is done by a hypernetwork where gradient from different agents upload their gradient to update this hypernetwork.

**Summary Of The Review:**

This paper is an application paper which deployed a personalized federated learning reinforcement learning in energy optimization situation. The technical novelty of algorithm itself is not strong which is acceptable for application paper. But the analysis and experiment on the simulation does not support the paper to become a high-quality application paper. More experiment, problem constrains and explanations for the simulation should be included in the paper.

---

> ### Author Response · Authors · 2022-11-18
> **3: Responses to Reviewer 3**
>
> We are grateful to Reviewer 3 for their comments about reproducibility and experiment detailsThe reviewer states that we don't give enough details about our environment, and we believe this may have contributed to a small misunderstanding: the reviewer seems to believe we are trying to change human behavior in response to price signals when we are actually only trying to manipulate automated battery charging/discharging policies. We appreciate this criticism, and have augmented the Methods section with more details about prosumer dynamics in our environment. Here is the added text: _By ensuring that each prosumer has the ability to transact with either the utility or the microgrid, we incentivize the microgrid to output prices that are better than the utility, guaranteeing a better experience for prosumers under this microgrid structure. One important simplification we have made is that we model human behavior as fixed in $\vec{e}\_b$ relative to the price signal; we do not expect humans to change behaviors (e.g., eating lunch at a different time to take advantage of cheaper energy prices). We only model how distributed batteries could be automatically controlled to maximize the prosumer's profit._
>
> The reviewer states, **"The weakness of the paper is experiment part is not strong enough to show advantage of their algorithm. The experiments show your algorithm is not always work, where only a selected scenario it outperformed baseline."** We politely disagree with this point, arguing that while the reviewer has correctly pointed out that there are regimes in which our method does not work so well, that (1) successful algorithms do not always work across all tasks (see [SAC](https://arxiv.org/pdf/1801.01290.pdf) which misses some tasks it tried) and (2) part of the strength of our paper is that we have performed a detailed ablation of environmental dynamics and have given a valid rationale, i.e. heterogeneity and size, where PFH performs well in, (3) even if it doesn't outperform certain regimes, it still performs competitively with other baselines, (4) even when it matches baseline performance, our algorithm still has a net positive effect because of the fact that we can transfer learning to new microgrids after training has converged, drastically reducing the amount of time necessary to achieve good performance on new microgrids. We argue that we have produced a nuanced analysis and have not suppressed failures, and that this is valuable knowledge and analysis that the community should be exposed to.

---

### Official Review · Reviewer_a54n · 2022-10-24

**Confidence:** 4
**Correctness:** 3
**Technical Novelty And Significance:** 2
**Empirical Novelty And Significance:** Not applicable
**Recommendation:** 3

**Clarity, Quality, Novelty And Reproducibility:**

Reproducibility: I appreciate the details of how the hyperparameters are selected are given in the appendix, but there seems no reproducibility statement in the current paper.

Clarity and Novelty: Please see the detailed reviews above.


**Strength And Weaknesses:**

Strength

1. This paper is the first to combine PFH with RL.

2. The experiment analyses are sufficient. The authors give detailed discussions on the possible reasons why their proposed method outperforms or is outperformed by baseline methods in different environments.

Weaknesses

1. Novelty: My greatest concern is about novelty. This paper seems to be a bit incremental in the sense that it directly combines the PFH with the proximal policy optimization (PPO) algorithm and then applies it in the specific energy pricing domain. The authors also do not seem to give any discussions about whether there are any technical difficulties when combining PFH with PPO and how they tackle these difficulties.

2. Technical Soundness: Though the proposed method applies the federated learning (FL) framework and trains all the personalized models without the communication of raw data, I still find it a bit overclaiming that the proposed method is for “privacy preservation”. As shown in Figure 1, the gradients of personalized models are sent to the central server. However, several works have shown that privacy leakage is still possible, even when only the gradients are communicated [1,2].

3. Experiments:

(a)  The proposed method indeed outperforms all the baseline methods when the heterogeneous tasks have medium diversity. However, all the experiments are conducted on synthetic datasets, making the results less convincing. It may be better if the authors can conduct more empirical evaluations on environments using real-world datasets.

(b)  The decentralized MARL method using networked agents also does not require the CTDE framework and can even be trained in a fully decentralized manner [6,7]. Combining with the differential privacy (DP) framework [8] to protect the information in the communication between networked agents, the decentralized MARL method is rigorously privacy-preserving, and can also be applied to the energy pricing domain. I would suggest the authors also include the DP-based privacy-preserving decentralized MARL method as a baseline to make the experiments more convincing.

(c)  Is there any criterion or clue about when the diversity is "medium"?

4. Literature Review: Several works have already shown the feasibility of combining FL with RL or MARL [3,4,5], which are missing in the current related work section. I would suggest the authors give more discussions and comparisons with these works.

5. Presentation: Overall, the presentation is clear but some parts of the paper can be further improved. For instance, before the method section, it would be better if a section introducing the setting or the preliminaries of the problem concerned in this paper is given (say, the basic preliminaries of RL). Besides, the sizes of some figures and tables in the experiment section can be reduced to put the related work section from the appendix to the main body of the paper for better readability. Also, it is unclear to me what Eq. (1) means, and it would be better if further explanations are given. For Eq. (2), it may be inappropriate to say that some RL agent aims to maximize “a reward”. In general, the objective of RL is to maximize the long-term expected total rewards instead of an instantaneous reward at some time step.

[1] Melis, L., Song, C., De Cristofaro, E., & Shmatikov, V. (2019, May). Exploiting unintended feature leakage in collaborative learning. In 2019 IEEE symposium on security and privacy (SP) (pp. 691-706). IEEE.

[2] Hitaj, B., Ateniese, G., & Perez-Cruz, F. (2017, October). Deep models under the GAN: information leakage from collaborative deep learning. In Proceedings of the 2017 ACM SIGSAC conference on computer and communications security (pp. 603-618).

[3] Xu, X., Li, R., Zhao, Z., & Zhang, H. (2021). The gradient convergence bound of federated multi-agent reinforcement learning with efficient communication. arXiv preprint arXiv:2103.13026.

[4] Kwon, D., Jeon, J., Park, S., Kim, J., & Cho, S. (2020). Multiagent DDPG-based deep learning for smart ocean federated learning IoT networks. IEEE Internet of Things Journal, 7(10), 9895-9903.

[5] Zhang, W., Yang, D., Wu, W., Peng, H., Zhang, N., Zhang, H., & Shen, X. (2021). Optimizing federated learning in distributed industrial IoT: A multi-agent approach. IEEE Journal on Selected Areas in Communications, 39(12), 3688-3703.

[6] Zhang, K., Yang, Z., & Başar, T. (2021). Decentralized multi-agent reinforcement learning with networked agents: Recent advances. Frontiers of Information Technology & Electronic Engineering, 22(6), 802-814.

[7] Zhang, K., Yang, Z., Liu, H., Zhang, T., & Basar, T. (2018, July). Fully decentralized multi-agent reinforcement learning with networked agents. In International Conference on Machine Learning (pp. 5872-5881). PMLR.

[8] Dwork, C., & Roth, A. (2014). The algorithmic foundations of differential privacy. Foundations and Trends® in Theoretical Computer Science, 9(3–4), 211-407.

**Summary Of The Paper:**

This paper makes the first step to combine personalized federated hypernetworks (PFH) with reinforcement learning (RL) and applies their method in the specific field of price-setting for energy demand response. The proposed method can work when the common centralized training with decentralized execution (CTDE) framework in multi-agent reinforcement learning (MARL) is not applicable. Empirically, this paper demonstrates that PFH enables efficient personalized model learning in multi-task MARL on experiments with synthetic datasets when the heterogeneous tasks have moderate diversity.

**Summary Of The Review:**

Though this paper makes the first step to combine PFH with PPO, the combination seems a bit incremental, and all the experiments are only conducted on the environments using synthetic datasets. Besides, the current literature review seems not sufficient. Therefore, I am afraid that this paper is not ready for publication as an ICLR paper.

---

> ### Author Response · Authors · 2022-11-18
> **2.1: Responses to Reviewer 2:**
>
> We are grateful to Reviewer 2 for their extremely in-depth and detailed feedback. Reviewer 2 states, **"the authors also do not seem to give any discussions about whether there are any technical difficulties when combining PFH with PPO and how they tackle these difficulties**". We appreciate this comment. We have linked the hyperparameter sweep as a solution to general RL hyperparameter sensitivity exacerbated by having to learn across heterogeneous environments, and expanded this in the text. We have included language pointing this out as a difficulty in combining PFH with PPO, as well as the issue of training on non-IID data in RL is exacerbated by non-IID data from working with different environments as well. Here is the added text: _We explain FedAvg's poor performance as follows: unlike supervised learning, RL requires exploration and already suffers from non-IID data. When aggregating learning across different heterogeneous environments, this issue of learning from non-IID data may have been exacerbated, slowing down learning. Furthermore, the setting of federated learning may have made the RL algorithm more sensitive to hyperparameters, as the set of hyperparameters that works for all tasks is likely smaller than the set that works for any one task. We conducted an extensive hyperparameter sweep that is documented in Appendix \ref{sec:hyper} to account for this issue with combining federated learning with RL. Meanwhile, the hypernetwork is able to learn how to build RL policies that are less sensitive to these hyperparameters because it outputs agents personalized to each task._
>
> Reviewer 2 states: **"I still find it a bit overclaiming that the proposed method is for "privacy preservation". As shown in Figure 1, the gradients of personalized models are sent to the central server. However, several works have shown that privacy leakage is still possible, even when only the gradients are communicated"** This is a really fair point that Reviewer 2 makes, but it is more a criticism of the whole field of FL than just our paper. We preserve privacy to the extent that other papers in the field do. Nevertheless, we appreciate this point, have added it as a disclaimer to the Related Works section, and have included the citations listed as well as additional citations from FL published previously in ICLR. We have added the following: _We note that ``privacy-preservation'' might be to an extent an overstatement, as works have shown that the transmission of gradients can allow one to recreate private data \citep{...}. Thus, while we note that our work guarantees privacy to the extent of other works within the field of federated learning \citep{...}, one should apply the term privacy-preservation with the same caveats to our work as to the rest of the field._
>
> The reviewer states, **"I would suggest the authors also include the DP-based privacy-preserving decentralized MARL method as a baseline to make the experiments more convincing."** We believe that the completely decentralized setting and federated learning setting may be too different from each other to be comparable, but we can try to do this if the TPC allows.

---

> ### Author Response · Authors · 2022-11-18
> **2.2: Responses to Reviewer 2:**
>
> The reviewer states, **"Is there any criterion or clue about when the diversity is "medium"?** Which we take to be a fair point about judging the range of diversity in environments. We had a good reason to determine the medium diversity: it was a mean and standard deviation which describes a distribution covering 95% of solar panel and battery sizes currently produced. Thus, it was natural in our environment. We: (1) have added in language explaining why we have judged medium diversity _("We note here that we have chosen the low, medium, and high cases such that 95\% of samples (i.e. 2 standard deviations around the mean) in the high case hit realistic bounds in the environment; i.e., 0 (an obvious lower bound) and 200\footnote{200 is a realistic upper bound in both solar panels and batteries: 200 solar panels would require an area of ~60 x 70 ft, which bounds the square footage of many commercial roofs, and 200 batteries would be a realistic upper bound of entities not engaging in commercial grid services.}.")_ (2) argue that a similar reasoning could be performed for other environments that the reader uses, (3) this shows a strength of our paper in being able to apply subject matter expertise in addition to an ML problem of merit.
>
> The reviewer states: **"Several works have already shown the feasibility of combining FL with RL or MARL [3,4,5], which are missing in the current related work section. I would suggest the authors give more discussions and comparisons with these works."** We have added these, thank you for suggesting this.
>
> The reviewer states **"Before the method section, it would be better if a section introducing the setting or the preliminaries of the problem concerned in this paper is given (say, the basic preliminaries of RL)."** We have done that, thank you. " **Besides, the sizes of some figures and tables in the experiment section can be reduced to put the related work section from the appendix to the main body of the paper for better readability."** We reduced the figures but had to add to the related works and other sections in order to address the valid points this review (and others) make. Thus, we were unable to move the Related works section up. "**Also, it is unclear to me what Eq. (1) means, and it would be better if further explanations are given."** Thank you, we have clarified, writing _The first term, i.e. the element-wise maximum, is thus the \textit{gross profit} from energy each prosumer sells, and the second term, i.e. the element-wise minimum, is the \textit{gross expenditures} from energy each prosumer buys_. **"For Eq. (2), it may be inappropriate to say that some RL agent aims to maximize "a reward". In general, the objective of RL is to maximize the long-term expected total rewards instead of an instantaneous reward at some time step."** Very fair point, thank you. We have clarified.

---

> > ### Comment · Reviewer_a54n · 2022-11-30
> > **Response to authors**
> >
> > Thanks for the detailed responses. I also have read reviews of other reviewers as well as the corresponding responses. However, it appears that my main concern regarding the novelty remains unresolved and I would like to maintain my score.

---

> > > ### Author Response · Authors · 2022-12-09
> > > **Response to Reviewer 2**
> > >
> > > Thank you for taking the time to read through our paper and responses/reviews. We addressed one of your concerns about novelty by including discussion about difficulties inherent in combing PFH with an RL algorithm like PPO, such as sensitivity to hyperparameters. We would also like to argue that the combination of PFH and RL alone, is a significant contribution to the field. PFH has only been demonstrated in supervised learning settings before, and RL is a different setting that is known to be much more difficult to solve due to hyperparameter sensitivity, lack of sample-efficiency, and the increased difficulty of training policies that can generalize to new environments. We addressed the first difficulty with an expansive hyperparameter search, and demonstrated how PFH helped the latter two difficulties through the few-shot transfer-learning experiment we highlighted in Response 0 above.

---

### Official Review · Reviewer_D68C · 2022-10-26

**Confidence:** 4
**Clarity, Quality, Novelty And Reproducibility:** Limited quality and novelty.
**Correctness:** 2
**Technical Novelty And Significance:** 1
**Empirical Novelty And Significance:** 1
**Recommendation:** 1

**Strength And Weaknesses:**

Weaknesses:
- The novelty of this work is limited. It seems that this work is a simple application of previous works. The problem being studied is also restricted (one specific environment). There's no evidence in the paper that the proposed method works for general RL problems.
- Significantly missing important related works in both FL / personalized FL / RL.
- The motivation of this work is unclear. What is the unique challenge in RL that prevents existing personalized FL framework to be deployed (for example, pFedMe [1], Ditto [2], etc.)

[1] T Dinh, C., Tran, N., & Nguyen, J. (2020). Personalized federated learning with moreau envelopes. Advances in Neural Information Processing Systems, 33, 21394-21405.

[2] Li, T., Hu, S., Beirami, A., & Smith, V. (2021, July). Ditto: Fair and robust federated learning through personalization. In International Conference on Machine Learning (pp. 6357-6368). PMLR.

**Summary Of The Paper:**

This work proposed applying hypernetwork for personalized federated reinforcement learning. Empirical results show that the proposed method achieves better performance compared to FedAvg.

**Summary Of The Review:**

This paper doesn't demonstrate enough novelty and contribution to the field of RL and FL. This is below the standard for ICLR.

---

> ### Author Response · Authors · 2022-11-18
> **Responses to Reviewer 1**
>
> We are grateful to Reviewer 1 for their insight into other personalized federated learning works, but we believe there may be some misunderstanding as to the novelty of our work. The reviewer implies that this work is not novel, but provides two references for personalized supervised learning only, even though our work is on personalized federated reinforcement learning. We tried to make clear that this was the first example of personalized federated RL that we could find, and we try to demonstrate this in our literature review. Reviewer 2 even notes, as a strength, that our work is the first application of PFH to RL.
>
> When Reviewer 1 breaks down the fields, they list: Federated Learning, Personalized Federated Learning, and Reinforcement Learning, excluding "Personalized Federated Reinforcement Learning", which proves the point that it's not an already existing category of RL strategies. All in all, we understand that this may seem like a trivial distinction to members outside the RL community, but RL is a different enough task to supervised learning that we (and Reviewer 2) believe that this distinction is an important one to make.
>
> Reviewer 1 also states that we are missing some important works in our literature review; we thank them for this comment and have added several new works on Personalized Federated Learning, Personalized Federated Reinforcement Learning, and Decentralized Multi-Agent Reinforcement Learning to our Related Works section.

---

### Author Response · Authors · 2022-11-18
**Rebuttal Format**

To the ICLR TPC and our generous reviewers:

We thank you for your time and effort in receiving, reading, and reviewing our work. We believe in the strength of our work, and in the process of honoring and responding to anonymous feedback, and so we choose to rebut the recommendations not to accept our manuscript. We do so in good faith, and hope that it helps to improve the process for all. First, we will note a general observation. Then, we will address each reviewers' points.

A brief note on formatting: **main points are in bold** , either by us or our reviewers. General commentary and response by us is in normal font. _Additions we make to our original manuscript in response to comments are italicized._

---

### Author Response · Authors · 2022-11-18
**0: Response(s) to points that more than one reviewer made, or didn't make**

The first thing we would like to note is that **none of the reviewers commented on our few-shot transfer learning experiment** , which we regard as one of the main contributions of this work. We showed that the PFH could very quickly adapt to a new set of microgrids – this has extreme implications for the feasibility of using PFH for this application, as it will cut down on data inefficiency as new microgrids join the system. It is also a new, novel application of PFH as the original paper did not demonstrate this capability.

Reviewers 1 and 2 note that **we do not demonstrate PFH across general environments**. We actually do test our architecture in 9 different tasks within our environment task suite, and so we have indeed tried to demonstrate some diversity in task. However, the reviewer's comment is not without reason in that we have not demonstrated other environment suites. In our initial planning of this work, we surveyed multi-agent benchmarks and found no environment that allowed us to control the heterogeneity of agent environments within the same task, something that we believed was critical to test. However, in investigating this comment, we believe we may be able to test this in a [Petting Zoo Multi-Particle Environmen](https://pettingzoo.farama.org/environments/mpe/)t, but it would require non-trivial software engineering, so we leave this for a future work.

Reviewers 2 and 3 stated that reproducibility of our work may be challenging. To remedy this, we have added the recommended reproducibility statement in the appendix, and uploaded some anonymized code.

---

### Decision · Program_Chairs · 2023-01-20

**Decision:**

Reject

**Justification For Why Not Higher Score:**

* Low novelty (straightforward combination of PFH and PPO)
* Weak experimental evaluation (only one environment suite)


**Justification For Why Not Lower Score:**

NA

**Metareview: Summary, Strengths And Weaknesses:**

The paper proposes an approach that combines personalilzed federated networks with RL

Strengths:
* Personalized federated RL is a problem that has not be explored extensively
* The applicatioh of energy demand response is nice

Weaknesses:
* Low novelty (straightforward combination of PFH and PPO)
* Weak experimental evaluation (only one environment suite)

Given the incremental nature of the work and weak experiments, this work is not ready for publication.